# LEARNING MULTI-SCALE LOCAL CONDITIONAL PROBABILITY MODELS OF IMAGES

**Zahra Kadkhodaie**
CDS, New York University
zk388@nyu.edu

**Florentin Guth**
DI, ENS, CNRS, PSL University
florentin.guth@ens.fr

**Stéphane Mallat**
Collège de France
Flatiron Institute, Simons Foundation
stephane.mallat@ens.fr

**Eero P. Simoncelli**
CNS, Courant, and CDS, New York University
Flatiron Institute, Simons Foundation
eero.simoncelli@nyu.edu

## ABSTRACT

Deep neural networks can learn powerful prior probability models for images, as evidenced by the high-quality generations obtained with recent score-based diffusion methods. But the means by which these networks capture complex global statistical structure, apparently without suffering from the curse of dimensionality, remain a mystery. To study this, we incorporate diffusion methods into a multi-scale decomposition, reducing dimensionality by assuming a stationary local Markov model for wavelet coefficients conditioned on coarser-scale coefficients. We instantiate this model using convolutional neural networks (CNNs) with local receptive fields, which enforce both the stationarity and Markov properties. Global structures are captured using a CNN with receptive fields covering the entire (but small) low-pass image. We test this model on a dataset of face images, which are highly non-stationary and contain large-scale geometric structures. Remarkably, denoising, super-resolution, and image synthesis results all demonstrate that these structures can be captured with significantly smaller conditioning neighborhoods than required by a Markov model implemented in the pixel domain. Our results show that score estimation for large complex images can be reduced to low-dimensional Markov conditional models across scales, alleviating the curse of dimensionality.

Deep neural networks (DNNs) have produced dramatic advances in synthesizing complex images and solving inverse problems, all of which rely (at least implicitly) on prior probability models. Of particular note is the recent development of "diffusion methods" (Sohl-Dickstein et al., 2015), in which a network trained for image denoising is incorporated into an iterative algorithm to draw samples from the prior (Song & Ermon, 2019; Ho et al., 2020; Song et al., 2021), or to solve inverse problems by sampling from the posterior (Kadkhodaie & Simoncelli, 2020; Cohen et al., 2021; Kawar et al., 2021; Daras et al., 2022). The prior in these procedures is implicitly defined by the learned denoising function, which depends on the prior through the score (the gradient of the log density). But density or score estimation is notoriously difficult for high-dimensional signals because of the curse of dimensionality: worst-case data requirements grow exponentially with the data dimension. How do neural network models manage to avoid this curse?

Traditionally, density estimation is made tractable by assuming simple low-dimensional models, or structural properties that allow factorization into products of such models. For example, the classical Gaussian spectral model for images or sounds rests on an assumption of translation-invariance (stationarity), which guarantees factorization in the Fourier domain. Markov random fields (Geman & Geman, 1984) assume localized conditional dependencies, which guarantees that the density can be factorized into terms acting on local, typically overlapping neighborhoods (Clifford & Hammersley, 1971). In the context of images, these models have been effective in capturing local properties, but are not sufficiently powerful to capture long-range dependencies. Multi-scale image decompositions offered a mathematical and algorithmic framework better suited for the structural properties of images (Burt & Adelson, 1983; Mallat, 2008). The multi-scale representation facilitates handling of larger

structures, and local (Markov) models have captured these probabilistically (e.g., Chambolle et al. (1998); Malfait & Roose (1997); Crouse et al. (1998); Buccigrossi & Simoncelli (1999); Paget & Longstaff (1998); Mihçak et al. (1999); Wainwright et al. (2001); Şendur & Selesnick (2002); Portilla et al. (2003); Cui & Wang (2005); Lyu & Simoncelli (2009)). Recent work, inspired by renormalization group theory in physics, has shown that probability distributions with long-range dependencies can be factorized as a product of Markov *conditional* probabilities over wavelet coefficients (Marchand et al., 2022). Although the performance of these models is eclipsed by recent DNN models, the concepts on which they rest—stationarity, locality and multi-scale conditioning—are still of fundamental importance. Here, we use these tools to constrain and study a score-based diffusion model.

A number of recent DNN image synthesis methods—including variational auto-encoders (Chen et al., 2018), generative adversarial networks (Gal et al., 2021) normalizing flow models (Yu et al., 2020; Li, 2021)), and diffusion models (Ho et al., 2022; Guth et al., 2022)—use coarse-to-fine strategies, generating a sequence of images of increasing resolution, each seeded by its predecessor. With the exception of the last, these do not make explicit the underlying conditional densities, and none impose locality restrictions on their computation. On the contrary, the stage-wise conditional sampling is typically accomplished with huge DNNs (up to billions of parameters), with global receptive fields.

Here, we develop a low-dimensional probability model for images decomposed into multi-scale wavelet sub-bands. Following the renormalization group approach, the image probability distribution is factorized as a product of conditional probabilities of its wavelet coefficients conditioned by coarser scale coefficients. We assume that these conditional probabilities are local and stationary, and hence can be captured with low-dimensional Markov models. Each conditional score can thus be estimated with a conditional CNN (cCNN) with a small receptive field (RF). The score of the coarse-scale low-pass band (a low-resolution version of the image) is modeled using a CNN with a global RF, enabling representation of large-scale image structures and organization. We test this model on a dataset of face images, which present a challenging example because of their global geometric structure. Using a coarse-to-fine anti-diffusion strategy for drawing samples from the posterior (Kadkhodaie & Simoncelli, 2021), we evaluate the model on denoising, super-resolution, and synthesis, and show that locality and stationarity assumptions hold for conditional RF sizes as small as $9 \times 9$ without harming performance. In comparison, the performance of CNNs restricted to a fixed RF size in the pixel domain dramatically degrades when the RF is reduced to such sizes. Thus, high-dimensional score estimation for images can be reduced to low-dimensional Markov conditional models, alleviating the curse of dimensionality. A software implementation is available at https://github.com/LabForComputationalVision/local-probability-models-of-images

# 1 Markov Wavelet Conditional Models

Images are high-dimensional vectors. Estimating an image probability distribution or its score therefore suffer from the curse of dimensionality, unless one limits the estimation to a relatively low-dimensional model class. This section introduces such a model class as a product of Markov conditional probabilities over multi-scale wavelet coefficients.

Markov random fields (Dobrushin, 1968; Sherrington & Kirkpatrick, 1975) define low-dimensional models by assuming that the probability distribution has local conditional dependencies over a graph, which is known a priori. One can then factorize the probability density into a product of conditional probabilities, each defined over a small number of variables (Clifford & Hammersley, 1971). Markov random fields have been used to model stationary texture images, with conditional dependencies within small spatial regions of the pixel lattice. At a location $u$, such a Markov model assumes that the pixel value $x(u)$, conditioned on pixel values $x(v)$ for $v$ in a neighborhood of $u$, is independent from all pixels outside this neighborhood. Beyond stationary textures, however, the chaining of short-range dependencies in pixel domain has proven insufficient to capture the complexity of long-range geometrical structures. Many variants of Markov models have been proposed (e.g., Geman & Geman (1984); Malfait & Roose (1997); Cui & Wang (2005)), but none have demonstrated performance comparable to recent deep networks while retaining a local dependency structure.

Based on the renormalization group approach in statistical physics (Wilson, 1971), new probability models are introduced in Marchand et al. (2022), structured as a product of probabilities of wavelet coefficients conditioned on coarser-scale values, with spatially local dependencies. These Markov

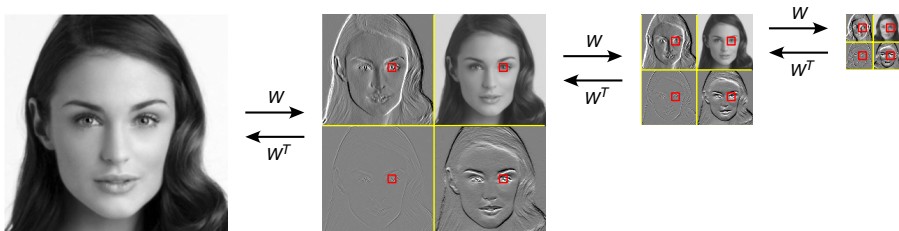

Figure 1: Markov wavelet conditional model structure. At each scale $j$, an orthogonal wavelet transform $W$ decomposes an image $\boldsymbol{x}_{j-1}$ into three wavelet channels, $\bar{\boldsymbol{x}}_j$, containing vertical, horizontal, and diagonal details, and a low-pass channel $\boldsymbol{x}_j$ containing a coarse approximation of the image, all subsampled by a factor of two. At each scale $j$, we assume a Markov wavelet conditional model, in which the probability distribution of any wavelet coefficient of $\bar{\boldsymbol{x}}_j$ (here, centered on the left eye), conditioned on values of $\boldsymbol{x}_j$ and $\bar{\boldsymbol{x}}_j$ in a local spatial neighborhood (red squares), is independent of all coefficients of $\bar{\boldsymbol{x}}_j$ outside this neighborhood.

conditional models have been applied to ergodic stationary physical fields, with simple conditional Gibbs energies that are parameterized linearly. Here, we generalize such models by parameterizing conditional Gibbs energy gradients with deep conditional convolutional neural networks having a local RF. This yields a class of Markov wavelet conditional models that can generate complex structured images, while explicitly relying on local dependencies to reduce the model dimensionality.

An orthonormal wavelet transform uses a convolutional and subsampling operator $W$ defined with conjugate mirror filters (Mallat, 2008), to iteratively compute wavelet coefficients (see Figure 1). Let $\boldsymbol{x}_0$ be an image of $N \times N$ pixels. For each scale $j > 1$, the operator $W$ decomposes $\boldsymbol{x}_{j-1}$ into:

$$W\boldsymbol{x}_{j-1} = (\bar{\boldsymbol{x}}_j, \boldsymbol{x}_j),$$

where $\boldsymbol{x}_j$ is a lower-resolution image and $\bar{\boldsymbol{x}}_j$ is an array of three wavelet coefficient images, each with dimensions $N/2^j \times N/2^j$, as illustrated in Figure 1. The inverse wavelet transform iteratively computes $\boldsymbol{x}_{j-1} = W^T(\bar{\boldsymbol{x}}_j, \boldsymbol{x}_j)$.

We now introduce the wavelet conditional factorization of probability models. Since $W$ is orthogonal, the probability density of $\boldsymbol{x}_{j-1}$ is also the joint density of $(\boldsymbol{x}_j, \bar{\boldsymbol{x}}_j)$. It can be factorized by conditioning on $\boldsymbol{x}_j$:

$$p(\boldsymbol{x}_{j-1}) = p(\boldsymbol{x}_j, \bar{\boldsymbol{x}}_j) = p(\boldsymbol{x}_j)p(\bar{\boldsymbol{x}}_j|\boldsymbol{x}_j).$$

This is performed $J$ times, so that the lowest resolution image $\boldsymbol{x}_J$ is small enough, which yields:

$$p(\boldsymbol{x}) = p(\boldsymbol{x}_J)\prod_{j=1}^{J} p(\bar{\boldsymbol{x}}_j|\boldsymbol{x}_j). \tag{1}$$

The conditional distributions $p(\bar{\boldsymbol{x}}_j|\boldsymbol{x}_j)$ specify the dependencies of image details at scale $j$ conditioned on the coarser scale values, and may be expressed in terms of a conditional Gibbs energy:

$$p(\bar{\boldsymbol{x}}_j|\boldsymbol{x}_j) = \mathcal{Z}_j(\boldsymbol{x}_j)^{-1} e^{-E_j(\bar{\boldsymbol{x}}_j|\boldsymbol{x}_j)}, \tag{2}$$

where $\mathcal{Z}(\boldsymbol{x}_j)$ is the normalization constant for each $\boldsymbol{x}_j$. The conditional Gibbs energies (2) have been used in the wavelet conditional renormalization group approach to obtain a stable parameterization of the probability model even at critical phase transitions, when the parameterization of the global Gibbs energy becomes singular (Marchand et al., 2022).

Local wavelet conditional renormalization group models (Marchand et al., 2022) further impose that $p(\bar{\boldsymbol{x}}_j|\boldsymbol{x}_j)$ is a conditional Markov random field. That is, the probability distribution of a wavelet coefficient of $\bar{\boldsymbol{x}}_j$ conditioned on values of $\boldsymbol{x}_j$ and $\bar{\boldsymbol{x}}_j$ in a restricted spatial neighborhood is independent of all coefficients of $\bar{\boldsymbol{x}}_j$ and $\bar{\boldsymbol{x}}$ outside this neighborhood (see Figure 1). The Hammersley-Clifford theorem states that this Markov property is equivalent to imposing that $E_j$ can be written as a sum of potentials, which only depends upon values of $\bar{\boldsymbol{x}}_j$ and $\boldsymbol{x}_j$ over local cliques (Clifford & Hammersley, 1971). This decomposition substantially alleviates the curse of dimensionality, since one only needs to estimate potentials over neighborhoods of a fixed size which does not grow with

the image size. To model ergodic stationary physical fields, the local potentials of the Gibbs energy $E_j$ have been parameterized linearly using physical models Marchand et al. (2022).

We generalize Markov wavelet conditional models by parameterizing the conditional score with a conditional CNN (cCNN) having small receptive fields (RFs):

$$-\nabla_{\bar{\boldsymbol{x}}_j} \log p(\bar{\boldsymbol{x}}_j|\boldsymbol{x}_j) = \nabla_{\bar{\boldsymbol{x}}_j} E_j(\bar{\boldsymbol{x}}_j|\boldsymbol{x}_j). \tag{3}$$

Computing the score (3) is equivalent to specifying the Gibbs model (2) without calculating the normalization constants $\mathcal{Z}(\boldsymbol{x}_j)$, since these are not needed for noise removal, super-resolution or image synthesis applications.

## 2 SCORE-BASED MARKOV WAVELET CONDITIONAL MODELS

Score-based diffusion models have produced impressive image generation results (e.g., Song et al. (2021); Ho et al. (2022); Rombach et al. (2022); Saharia et al. (2022); Ramesh et al. (2022)). To capture large-scale properties, however, these networks require RFs that encompass the entire image. Our score-based wavelet conditional model leverages the Markov assumption to compute the score using cCNNs with small receptive fields, offering a low-dimensional parameterization of the image distribution while retaining long-range geometric structures.

Let $\boldsymbol{y} = \boldsymbol{x} + \boldsymbol{z}$ be a noisy observation of clean image $\boldsymbol{x} \in \mathbb{R}^{N \times N}$ drawn from $p(\boldsymbol{x})$, with $\boldsymbol{z} \sim \mathcal{N}(0, \sigma^2 \mathrm{Id})$ a sample of Gaussian white noise. The minimum mean squared error (MMSE) estimate of the true image is well-known to be the conditional mean of the posterior:

$$\hat{\boldsymbol{x}}(\boldsymbol{y}) = \int \boldsymbol{x} p(\boldsymbol{x}|\boldsymbol{y}) \mathrm{d}\boldsymbol{x}. \tag{4}$$

This integral can be re-expressed in terms of the score:

$$\hat{\boldsymbol{x}}(\boldsymbol{y}) = \boldsymbol{y} + \sigma^2 \nabla_{\boldsymbol{y}} \log p(\boldsymbol{y}). \tag{5}$$

This remarkable result, published in Miyasawa (1961), exposes a direct and explicit relationship between the score of probability distributions and denoising (we reproduce the proof in Appendix B for completeness). Note that the relevant density is not the image distribution, $p(\boldsymbol{x})$, but the *noisy observation density* $p(\boldsymbol{y})$. This density converges to $p(\boldsymbol{x})$ as the noise variance $\sigma^2$ goes to zero.

Given this relationship, the score can be approximated with a parametric mapping optimized to estimate the denoised image, $f(\boldsymbol{y}) \approx \hat{\boldsymbol{x}}(\boldsymbol{y})$. Specifically, we implement this mapping with a CNN, and optimize its parameters by minimizing the denoising squared error $||f(\boldsymbol{y}) - \boldsymbol{x}||^2$ over a large training set of images and their noise-corrupted counterparts. Given eq. (5), the denoising residual, $f(\boldsymbol{y}) - \boldsymbol{y}$, provides an approximation of the variance-weighted score, $\sigma^2 \nabla_{\boldsymbol{y}} \log p(\boldsymbol{y})$. Also known as denoising score matching (Vincent, 2011), such denoiser-estimated score functions have been used in iterative algorithms for drawing samples from the density (Song et al., 2021; Ho et al., 2020; Dhariwal & Nichol, 2021; Ho et al., 2022), or solving inverse problems (Kadkhodaie & Simoncelli, 2020; Cohen et al., 2021; Kawar et al., 2021; Laumont et al., 2022).

To model the conditional wavelet distribution $p(\bar{\boldsymbol{x}}_j|\boldsymbol{x}_j)$, Guth et al. (2022) parameterize the score $\nabla_{\bar{\boldsymbol{y}}_j} \log p(\bar{\boldsymbol{y}}_j|\boldsymbol{x}_j)$ of noisy wavelet coefficients $\bar{\boldsymbol{y}}_j$ conditioned on a clean low-pass image $\boldsymbol{x}_j$ with a cCNN (eq. (3)). Specifically, the cCNN takes as input three noisy wavelet detail channels, along with the corresponding low-pass channel, and generates three estimated detail channels. The network is trained to minimize mean square distance between $\bar{\boldsymbol{x}}_j$ and $f_j(\bar{\boldsymbol{y}}_j, \boldsymbol{x}_j)$. Thanks to a conditional extension of eq. (5), an optimal network computes $f_j(\bar{\boldsymbol{y}}_j, \boldsymbol{x}_j) = \nabla_{\bar{\boldsymbol{y}}_j} \log p(\bar{\boldsymbol{y}}_j|\boldsymbol{x}_j)$. Additionally, at the coarsest scale $J$, a CNN denoiser $f_J(\boldsymbol{y}_J)$ is trained to estimate the score of the low-pass band, $\nabla_{\boldsymbol{y}_J} \log p(\boldsymbol{x}_J)$ by minimizing mean square distance between $\boldsymbol{x}_J$ and $f_J(\boldsymbol{y}_J)$.

The following theorem proves that the Markov wavelet conditional property is equivalent to imposing that the cCNN RFs are restricted to the conditioning neighborhoods. The RF of a given element of the network response is defined as the set of input image pixels on which this element depends.

**Theorem 1.** *The wavelet conditional density $p(\bar{\boldsymbol{x}}_j|\boldsymbol{x}_j)$ is Markovian over a family of conditioning neighborhoods if and only if the conditional score $\nabla_{\bar{\boldsymbol{x}}_j} \log p(\bar{\boldsymbol{x}}_j|\boldsymbol{x}_j)$ can be computed with a network whose RFs are included in these conditioning neighborhoods.*

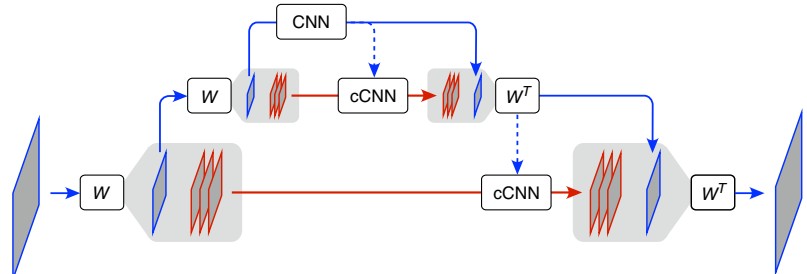

Figure 2: Wavelet conditional denoiser architecture used to estimate the score (illustrated for a two-scale decomposition). The input noisy image $\boldsymbol{y}$ (lower left) is decomposed by recursive application of a fast orthogonal wavelet transform $W$ into successive low-pass images $\boldsymbol{y}_j$ (blue) and three wavelet detail images $\bar{\boldsymbol{y}}_j$ (red). The coarsest low-pass image $\boldsymbol{y}_J$ is denoised using a CNN with a global receptive field to estimate $\hat{\boldsymbol{x}}_J$. At all other scales, a local conditional CNN (cCNN) estimates $\hat{\bar{\boldsymbol{x}}}_j$ from $\bar{\boldsymbol{y}}_j$ conditioned on $\hat{\boldsymbol{x}}_j$, from which $W^T$ recovers $\hat{\boldsymbol{x}}_{j-1}$.

The proof of the theorem is provided in Appendix A. Note that even if the conditional distribution of clean wavelet coefficients $p(\bar{\boldsymbol{x}}|\boldsymbol{x}_j)$ satisfies a local Markov property, the noisy distribution $p(\bar{\boldsymbol{y}}_j|\boldsymbol{x}_j)$ is in general not Markovian. However, we shall parameterize the scores with a cCNN with small RFs and hence show that both the noisy and clean distributions are Markovian. At each scale $1 \le j \le J$, the cCNN has RFs that are localized in both $\bar{\boldsymbol{y}}_j$ and $\boldsymbol{x}_j$, and have a fixed size over all scales, independent of the original image size. From Theorem 1, this defines the Markov conditioning neighborhoods of the learned model. The effect of the RF size is examined in the numerical experiments of Section 3.

Parameterizing the score with a convolutional network further implies that the conditional probability $p(\bar{\boldsymbol{x}}_j|\boldsymbol{x}_j)$ is stationary on the wavelet sampling lattice at scale $j$. Despite these strong simplifications, we shall see that these models models are able to capture complex long-range image dependencies in highly non-stationary image ensembles such as centered faces. This relies on the low-pass CNN, whose RF is designed to cover the entire image $\boldsymbol{x}_J$, and thus does not enforce local Markov conditioning nor stationarity. The product density of eq. (1) is therefore not stationary.

## 3 MARKOV WAVELET CONDITIONAL DENOISING

We now evaluate our Markov wavelet conditional model on a denoising task. We use the trained CNNs to define a multi-scale denoising architecture, illustrated in Figure 2. The wavelet transform of the input noisy image $\boldsymbol{y}$ is computed up to a coarse-scale $J$. The coarsest scale image is denoised by applying the denoising CNN learned previously: $\hat{\boldsymbol{x}}_J = f_J(\boldsymbol{y}_J)$. Then for $J \ge j \ge 1$, we compute the denoised wavelet coefficients conditioned on the previously estimated coarse image: $\hat{\bar{\boldsymbol{x}}}_j = f(\bar{\boldsymbol{y}}_j, \hat{\boldsymbol{x}}_j)$. We then recover a denoised image at the next finer scale by applying an inverse wavelet transform: $\hat{\boldsymbol{x}}_{j-1} = W^T(\hat{\bar{\boldsymbol{x}}}_j, \hat{\boldsymbol{x}}_j)$. At the finest scale we obtain the denoised image $\hat{\boldsymbol{x}} = \hat{\boldsymbol{x}}_0$.

Because of the orthogonality of $W$, the global MSE can be decomposed into a sum of wavelet MSEs at each scale, plus the coarsest scale error: $\|\boldsymbol{x} - \hat{\boldsymbol{x}}\|^2 = \sum_{j=1}^{J-1} \|\bar{\boldsymbol{x}}_j - \hat{\bar{\boldsymbol{x}}}_j\|^2 + \|\boldsymbol{x}_J - \hat{\boldsymbol{x}}_J\|^2$. The global MSE thus summarizes the precision of the score models computed over all scales. We evaluate the peak signal-to-noise ratio (PSNR) of the denoised image as a function of the noise level, expressed as the PSNR of the noisy image. We use the CelebA dataset (Liu et al., 2015) at $160 \times 160$ resolution. We use the simplest of all orthogonal wavelet decompositions, the Haar wavelet, constructed from separable filters that compute averages and differences of adjacent pairs of pixel values (Haar, 1910). All denoisers are "universal" (they can operate on images contaminated with noise of any standard deviation), and "blind" (they are not informed of the noise level). They all have the same depth and layer widths, and their receptive field size is controlled by changing the convolutional kernel size of each layer. Appendix C provides architecture and training details.

Figure 3 shows that the multi-scale denoiser based on a conditional wavelet Markov model outperforms a conventional denoiser that implements a Markov probability model in the pixel domain. More precisely, we observe that when the Markov structure is defined over image pixels, the performance degrades considerably with smaller RFs (Figure 3, left panel), especially at large noise

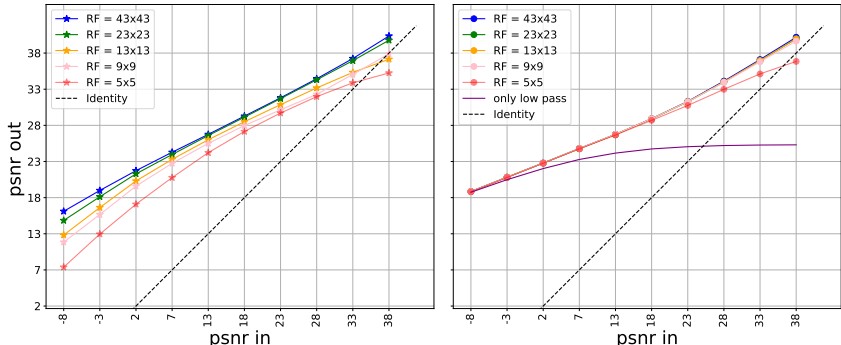

Figure 3: Comparison of denoiser performance on $160 \times 160$ test images from the CelebA dataset. Each panel shows the error of the denoised image as a function of the noise level, both expressed as the peak signal-to-noise ratio (PSNR). **Left:** Conventional CNN denoisers with different RF sizes. The blue curve shows performance of BF-CNN (Mohan* et al., 2020). The rest are BF-CNN variants with smaller RF obtained from setting some intermediate filter sizes to $1 \times 1$. **Right:** Multi-scale denoisers, as depicted in Figure 2, with different cCNN RF sizes. Note that the low-pass denoiser RF is $40 \times 40$ in all cases, and thus covers the entire low-pass band.

levels (low PSNR). Images thus contain long-range global dependencies that cannot be captured by Markov models that are localized within the pixel lattice. On the other hand, multi-scale denoising performance remains nearly the same for RF sizes down to $9 \times 9$, and degrades for $5 \times 5$ RFs only at small noise levels (high PSNR) (Figure 3, right panel). This is remarkable considering that, for the finest scale, the $9 \times 9$ RF implies conditioning on one percent of the coefficients. The wavelet conditional score model thus successfully captures long-range image dependencies, even with small Markov neighborhoods.

It is also worth noting that in the large noise regime (i.e., low PSNR), all multi-scale denoisers (even with RF as small as $5 \times 5$) significantly outperforms the conventional denoiser with the largest tested RF size ($43 \times 43$). The dependency on RF size in this regime demonstrates the inadequacy of local modeling in the pixel domain. On the contrary, the effective neighborhoods of the multi-scale denoiser are spatially global, but operate with spatially-varying resolution. Specifically, neighborhoods are of fixed size at each scale, but due to the subsampling, cover larger proportions of the image at coarser scales. The CNN applied to the coarsest low-pass band (scale $J$) is spatially global, and the denoising of this band alone explains the performance at the highest noise levels (magenta curve, Figure 3).

To further illustrate this point, consider the denoising examples shown in Figure 4. Since all denoisers are bias-free, they are piecewise linear (as opposed to piecewise affine), providing some interpretability (Mohan* et al., 2020). Specifically, each denoised pixel is computed as an adaptively weighted sum over the noise input pixels. The last panels show the equivalent adaptive linear filter that was used to estimate the pixel marked by the green square, which can be estimated from the associated row of the Jacobian. The top row shows denoising results of a conventional CNN denoiser for small images that are the size of the network RF. Despite very heavy noise levels, the denoiser exploits the global structure of the image, and produces a result approximating the clean image. The second row shows the results after training the same denoiser architecture on much larger images. Now the adaptive filter is much smaller than the image, and the denoiser solution fails to capture the global structure of the face. Finally, the last row shows that the multi-scale wavelet conditional denoiser can successfully approximate the global structure of a face despite the extreme levels of noise. Removing high levels of noise requires knowledge of global structure. In our multi-scale conditional denoiser, this is achieved by setting the RF size of the low-pass denoiser equal to the entire low-pass image size, similarly to the denoiser shown on the top row. Then, each successive conditioning stage provides information at a finer resolution, over ever smaller RFs relative to the coefficient lattice. The adaptive filter shown in the last column has a *foveated* structure: the estimate of the marked pixel depends on all pixels in the noisy image, but those that are farther away are only included within averages over larger blocks. Thus, imposing locality in the wavelet domain lifts the curse of dimensionality without loss of performance, as opposed to a locality (Markov) assumption in the pixel domain.

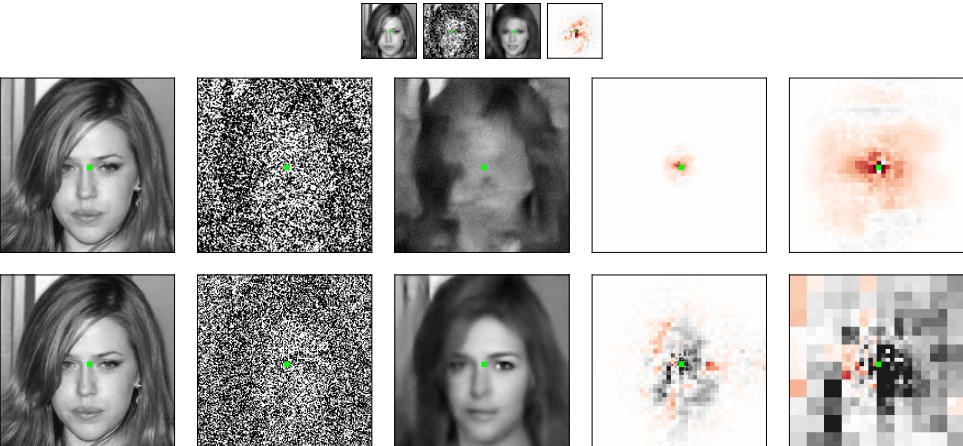

Figure 4: Denoising examples. Each row shows clean image, noisy image, denoised image, and the adaptive filter (one row of the Jacobian of the end-to-end denoising transformation) used by the denoiser to estimate a specific pixel, indicated in green. The heat-map ranges from red for most negative to black for most positive values. In the last two rows, the last column shows an enlargement of this adaptive filter, for enhanced visibility. Images are displayed proportional to their sizes. **Top row:** $40 \times 40$ images estimated with a CNN denoiser with RF $40 \times 40$. **Second row:** $160 \times 160$ images estimated with a CNN denoiser with RF $43 \times 43$. **Third row:** $160 \times 160$ images, estimated with the proposed conditional multi-scale denoiser of Figure 2. The denoiser uses a $40 \times 40$ RF for the coarsest scale, and $13 \times 13$ RFs for conditional denoising of subsequent finer scales.

## 4  MARKOV WAVELET CONDITIONAL SUPER-RESOLUTION AND SYNTHESIS

We generate samples from the learned wavelet conditional distributions in order to visually assess the quality of the model in a super-resolution task. We compare this approach with solving the super-resolution inverse problem directly using a CNN denoiser operating in the pixel domain. We also compare the models on image synthesis.

We first give a high-level description of our conditional generation algorithm. The low-resolution image $x_J$ is used to conditionally generate wavelet coefficients $\bar{x}_J$ from the conditional distribution $p(\bar{x}_J|x_J)$. An inverse wavelet transform next recovers a higher-resolution image $x_{J-1}$ from both $x_J$ and $\bar{x}_J$. The conditional generation and wavelet reconstruction are repeated $J$ times, increasing the resolution of the sample at each step. In the end, we obtain a sample $x$ from the full-resolution image distribution conditioned on the starting low-resolution image $p(x|x_J)$. $x$ is thus a stochastic super-resolution estimate of $x_J$.

To draw samples from the distributions $p(\bar{x}_j|x_j)$ implicitly embedded in the wavelet conditional denoisers, we use the algorithm of **?**, which performs stochastic gradient ascent on the log-probability obtained from the cCNN using eq. (5). This is similar to score-based diffusion algorithms (Song & Ermon, 2019; Ho et al., 2020; Song et al., 2021), but the timestep hyper-parameters require essentially no tuning, since stepsizes are automatically obtained from the magnitude of the estimated score. Extension to the conditional case is straightforward. The sampling algorithm is detailed in Appendix D. All the cCCN denoisers have a RF size of $13 \times 13$. Train and test images are from the CelebA HQ dataset (Karras et al., 2018) and of size $320 \times 320$. Samples drawn using the conditional denoiser correspond to a Markov conditional distribution with neighborhoods restricted to the RFs of the denoiser. We compare these with samples from a model with a local Markov neighbhorhood in the pixel domain. This is done using a CNN with a $40 \times 40$ RF trained to denoise full-resolution images, which approximates the score $\nabla \log p(x)$. Given the same low-pass image $x_J$, we can generate samples from $p(x|x_J)$ by viewing this as sampling from the image distribution of $x$ constrained by a linear measurements $x_J$. This is done with the same sampling algorithm, with a small modification, again described in Appendix D.

Figure 5 shows super-resolution samples from these two learned image models. The local Markov model in the pixel domain generates details that are sharp but artifactual and incoherent over long

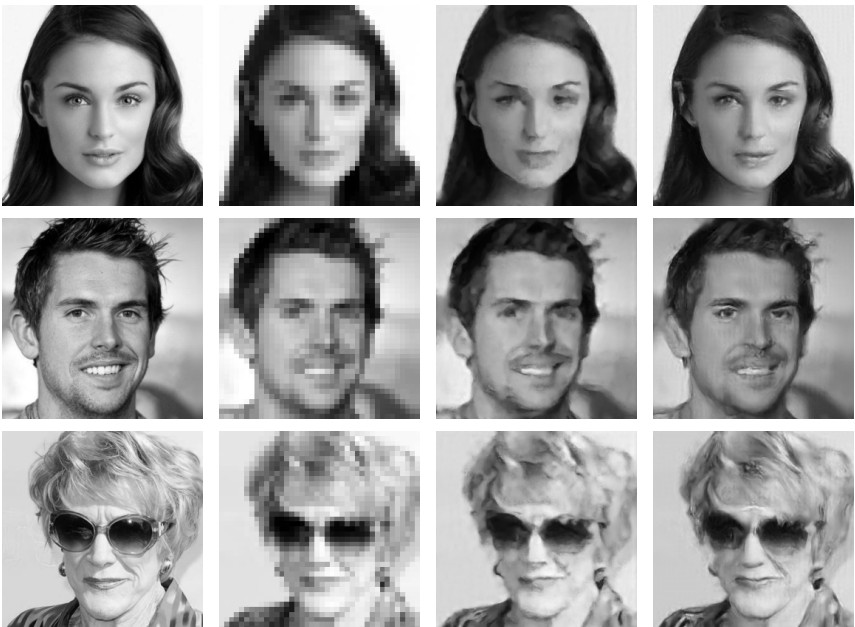

Figure 5: Super-resolution examples. **Column 1**: original images ($320 \times 320$). **Column 2**: Low-pass image of a 3-stage wavelet decomposition (downsampled to $40 \times 40$) expanded to full size for viewing. **Column 3**: Conditional generation of full-resolution images using CNN denoiser with RF of size $43 \times 43$. **Column 4**: Coarse-to-fine conditional generation using the multi-scale cCNN denoisers, each with RFs of size $13 \times 13$. Additional examples are shown in Appendix E.

spatial distances. On the other hand, the Markov wavelet conditional model produces much more natural-looking face images. This demonstrates the validity of our model: although these face images are not stationary (they have global structures shared across the dataset), and are not Markov in the pixel domain (there are clearly long-range dependencies that operate across the image), the *details* can be captured with local stationary Markov wavelet conditional distributions.

We also evaluated the Markov wavelet conditional model on image synthesis. We first synthesize a $40 \times 40$ terminal low-pass image using the score, $\nabla_{\boldsymbol{x}_J} \log p(\boldsymbol{x}_J)$, obtained from the low-pass CNN denoiser with a global RF. Again, unlike the conditional wavelet stages, this architectural choice does not enforce any local Markov structure nor stationarity. This global RF allows capturing global non-stationary structures, such as the overall face shape. The synthesis then proceeds using the same coarse-to-fine steps as used for super-resolution: wavelet coefficients at each successive scale are generated by drawing a sample using the cCNN conditioned on the previous scale.

The first (smallest) image in Figure 6 is generated from the low-pass CNN (see Appendix D for algorithm). We can see that it successfully captures the coarse structure of a face. This image is then refined by application of successive stages of the multi-scale super-resolution synthesis algorithm described above. The next three images in Figure 6 show successively higher resolution images generated in the procedure. For comparison, the last image in Figure 6 shows a sample generated using a conventional CNN with equivalent RFs trained on large face images. Once again, this illustrates that assuming spatially localized Markov property on the pixel lattice and estimating the score with a CNN with RF smaller than the image fails to capture the non-stationary distribution of faces. Specifically, the model is unable to generate structures larger than the RF size, and the samples are texture-like and composed of local regions resembling face parts.

We note that the quality of the generated images is not on par with the most recent score-based diffusion methods (e.g., Ramesh et al. (2022); Saharia et al. (2022); Rombach et al. (2022)), which have also been used for iterative super-resolution of an initial coarse-scale sample. These methods use much larger networks (more than a billion parameters, compared to ours which uses 600k parameters for the low-pass CNN and 200k for the cCNNs), and each scale-wise denoiser is itself a U-Net, with associated RF covering the entire image. Thus, the implicit probability models in these networks are

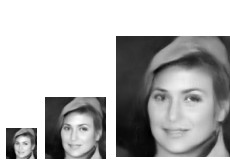 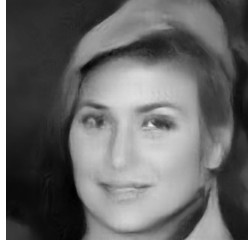 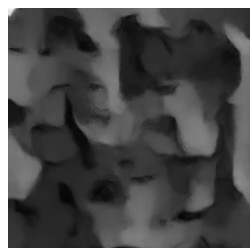

Figure 6: Image synthesis. **Left four images:** Coarse-to-fine synthesis, achieved by sampling the score learned for each successive conditional distribution. Synthesized images are shown at four resolutions, from coarse-scale only (leftmost, $40 \times 40$) to the finest scale (rightmost, $320 \times 320$). Conditional RFs are all $13 \times 13$. **Right image:** Synthesis using a pixel-domain CNN with a receptive field ($40 \times 40$) smaller than the synthesized image $320 \times 320$.

global, and it is an open question whether and how these architectures are able to escape the curse of dimensionality. Our local conditional Markov assumptions provide a step towards the goal of making explicit the probability model and its factorization into low-dimensional components.

## 5 DISCUSSION

We have generalized a Markov wavelet conditional probability model of image distributions, and developed an explicit implementation using cCNNs to estimate the conditional model scores. The resulting conditional wavelet distributions are stationary and Markov over neighborhoods corresponding to the cCNN receptive fields. The coarse-scale low-pass band is modeled using the score estimated with a CNN with global receptive fields. We trained this model on a dataset of face images, which are non-stationary with large-scale geometric features. We find that the model, even with relatively small cCNN RFs, succeeds in capturing these features, producing high-quality results on denoising and super-resolution tasks. We contrast this with local Markov models in the pixel domain, which are not able to capture these features, and are instead limited to stationary ergodic textures.

The Markov wavelet conditional model demonstrates that probability distributions of images can be factorized as products of conditional distributions that are local. This model provides a mathematical justification which can partly explain the success of coarse-to-fine diffusion synthesis (Ho et al., 2020; Guth et al., 2022), which also compute conditional scores at each scale. Although we set out to understand how factorization of density models could allow them to avoid the curse of dimensionality in training, it is worth noting that the dimensionality of the conditioning neighborhoods in our network is still uncomfortably high ($4 \times 9 \times 9 = 324$). Although this is reduced by a factor of roughly 300 relative to the image size ($320 \times 320 = 102,400$), and this dimensionality remains constant even if this image size is increased, it is still not sufficient to explain how the conditional score can be trained with realistic amounts of data. In addition, the terminal low-pass CNN operates globally (dimensionality $40 \times 40 = 1600$). Thus, the question of how to further reduce the overall dimensionality of the model remains open.

Our experiments were performed on cropped and centered face images, which present a particular challenge given their obvious non-stationarity. The conditional models are approximately stationary due to the fully convolutional structure of the cCNN operations (although this is partially violated by zero-padded boundary handling). As such, the non-stationary capabilities of the full model arise primarily from the terminal low-pass CNN, which uses spatially global RFs. We speculate that for more diverse image datasets (e.g., a large set of natural images), a much larger capacity low-pass CNN will be needed to capture global structures. This is consistent with current deep networks that generate high-quality synthetic images using extremely large networks (Ramesh et al., 2022; Saharia et al., 2022; Rombach et al., 2022). On the other hand, the cCNNs in our model all share the same architecture and local RFs, and may (empirically) be capturing similar local conditional structure at each scale. Forcing these conditional densities to be the same at each scale (through weight sharing of the corresponding cCNNs) would impose a scale invariance assumption on the overall model. This would further reduce model complexity, and enable synthesis and inference on images of size well beyond that of the training set.

## ACKNOWLEDGMENTS

We gratefully acknowledge the support and computing resources of the Flatiron Institute (a research division of the Simons Foundation), NSF NRT HDR Award 1922658 to the Center for Data Science at NYU, and a grant from the PRAIRIE 3IA Institute of the French ANR-19-P3IA-0001 program.

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

## A    PROOF OF THEOREM 1

To simplify notation, we drop the $j$ subscript. Let $I$ (resp. $J$) denote the set of indices of pixel values of $\bar{\boldsymbol{x}}$ (resp. $\boldsymbol{x}$). If $S$ is a set of indices, we denote $\bar{\boldsymbol{x}}(S) = (\bar{\boldsymbol{x}}(i))_{i \in S \cap I}$. Let $G$ be a graph whose nodes are $I \cup J$. For each $i \in I$, let $N(i) \subseteq I \cup J$ be the neighborhood of node $i$, with $i \notin N(i)$, and $N_+(i) = N(i) \cup \{i\}$.

To prove Theorem 1, we need to show that the local Markov property:

$$\forall i \in I, \ p\big(\bar{\boldsymbol{x}}(i) \,\big|\, \bar{\boldsymbol{x}}(I \setminus \{i\}), \boldsymbol{x}\big) = p\big(\bar{\boldsymbol{x}}(i) \,\big|\, \bar{\boldsymbol{x}}(N(i)), \boldsymbol{x}(N_+(i))\big), \tag{6}$$

is equivalent to the conditional score being computable with RFs restricted to neighborhoods:

$$\forall i \in I, \ \frac{\partial \log p}{\partial \bar{\boldsymbol{x}}(i)}\big(\bar{\boldsymbol{x}} \,\big|\, \boldsymbol{x}\big) = f_i\big(\bar{\boldsymbol{x}}(N_+(i)), \boldsymbol{x}(N_+(i))\big), \tag{7}$$

for some functions $f_i$.

We first prove that eq. (6) implies eq. (7). Let $i \in I$. We have the following factorization of the probability distribution:

$$
\begin{aligned}
p\big(\bar{\boldsymbol{x}} \,\big|\, \boldsymbol{x}\big) &= p\big(\bar{\boldsymbol{x}}(i) \,\big|\, \bar{\boldsymbol{x}}(I \setminus \{i\}), \boldsymbol{x}\big) \, p\big(\bar{\boldsymbol{x}}(I \setminus \{i\}) \,\big|\, \boldsymbol{x}\big) \\
&= p\big(\bar{\boldsymbol{x}}(i) \,\big|\, \bar{\boldsymbol{x}}(N(i)), \boldsymbol{x}(N_+(i))\big) \, p\big(\bar{\boldsymbol{x}}(I \setminus \{i\}) \,\big|\, \boldsymbol{x}\big),
\end{aligned}
$$

where we have used eq. (6) in the last step. Then, taking the logarithm and differentiating, only the first term remains:

$$\frac{\partial \log p}{\partial \bar{\boldsymbol{x}}(i)}\big(\bar{\boldsymbol{x}} \,\big|\, \boldsymbol{x}\big) = \frac{\partial \log p}{\partial \bar{\boldsymbol{x}}(i)}\big(\bar{\boldsymbol{x}}(i) \,\big|\, \bar{\boldsymbol{x}}(N(i)), \boldsymbol{x}(N_+(i))\big),$$

which proves eq. (7).

Reciprocally, we now prove that eq. (7) implies eq. (6). Let $i \in I$, and $\boldsymbol{\delta}_i(j) = \delta_{ij}$ where $\delta_{ij}$ is the Kronecker delta. We have, by integrating the partial derivative:

$$\log p\big(\bar{\boldsymbol{x}} \,\big|\, \boldsymbol{x}\big) = \log p\big(\bar{\boldsymbol{x}} - \bar{\boldsymbol{x}}(i)\boldsymbol{\delta}_i \,\big|\, \boldsymbol{x}\big) - \int_0^1 \frac{\partial \log p}{\partial \bar{\boldsymbol{x}}(i)}\big(\bar{\boldsymbol{x}} - t\bar{\boldsymbol{x}}(i)\boldsymbol{\delta}_i \,\big|\, \boldsymbol{x}\big)\mathrm{d}t$$

$$= \log p\big(\bar{\boldsymbol{x}} - \bar{\boldsymbol{x}}(i)\boldsymbol{\delta}_i \,\big|\, \boldsymbol{x}\big) - \int_0^1 f_i\big(\bar{\boldsymbol{x}}(N_+(i)) - t\bar{\boldsymbol{x}}(i)\boldsymbol{\delta}_i, \boldsymbol{x}(N_+(i))\big)\mathrm{d}t,$$

where we have used eq. (7) in the last step. Note that the first term does not depend on $\bar{\boldsymbol{x}}(i)$, while the second term only depends on $\bar{\boldsymbol{x}}(N_+(i))$ and $\boldsymbol{x}(N_+(i))$. This implies that when we condition on $\bar{\boldsymbol{x}}(N(i))$ and $\boldsymbol{x}(N_+(i))$, the density factorizes as a term which only involves $\bar{\boldsymbol{x}}(i)$ and a term which does not involve $\bar{\boldsymbol{x}}(i)$. This further implies conditional independence and thus eq. (6).

## B    PROOF OF EQ. (5)

Miyasawa's remarkable result (Miyasawa, 1961), sometimes attributed to Tweedie (as communicated by (Robbins, 1956)), is simple to prove (Raphan & Simoncelli, 2007). The observation distribution, $p(\boldsymbol{y})$ is obtained by marginalizing $p(\boldsymbol{y}, \boldsymbol{x})$:

$$p(\boldsymbol{y}) = \int p(\boldsymbol{y}|\boldsymbol{x})p(\boldsymbol{x})\mathrm{d}\boldsymbol{x} = \int g(\boldsymbol{y} - \boldsymbol{x})p(\boldsymbol{x})\mathrm{d}\boldsymbol{x},$$

where the noise distribution $g(\boldsymbol{z})$ is Gaussian. The gradient of the observation density is then:

$$\nabla_{\boldsymbol{y}}\, p(\boldsymbol{y}) = \frac{1}{\sigma^2}\int(\boldsymbol{x} - \boldsymbol{y})g(\boldsymbol{y} - \boldsymbol{x})p(\boldsymbol{x})\mathrm{d}\boldsymbol{x} = \frac{1}{\sigma^2}\int(\boldsymbol{x} - \boldsymbol{y})p(\boldsymbol{y}, \boldsymbol{x})\mathrm{d}\boldsymbol{x}.$$

Multiplying both sides by $\sigma^2/p(\boldsymbol{y})$ and separating the right side into two terms gives:

$$\sigma^2\frac{\nabla_{\boldsymbol{y}}\, p(\boldsymbol{y})}{p(\boldsymbol{y})} = \int \boldsymbol{x}p(\boldsymbol{x}|\boldsymbol{y})\mathrm{d}\boldsymbol{x} - \int \boldsymbol{y}p(\boldsymbol{x}|\boldsymbol{y})\mathrm{d}\boldsymbol{x} = \hat{\boldsymbol{x}}(\boldsymbol{y}) - \boldsymbol{y}.$$

## C    TRAINING AND ARCHITECTURE DETAILS

**Architecture**. The terminal low-pass CNN and all cCNNs are "bias-free": we remove all additive constants from convolution and batch-normalization operations (i.e., the batch normalization does not subtract the mean) (Mohan* et al., 2020). All networks contain 21 convolutional layers with no subsampling, each consisting of 64 channels. Each layer, except for the first and the last, is followed by a ReLU non-linearity and bias-free batch-normalization. Thus, the transformation is both homogeneous (of order 1) and translation-invariant (apart from handling of boundaries), at each scale. All convolutional kernels in the low-pass CNN are of size $3 \times 3$, resulting in a $43 \times 43$ RF size and $665,856$ parameters in total. Convolutional kernels in the cCNNs are adjusted to achieve different RF sizes. For example, a $13 \times 13$ RF arises from choosing $3 \times 3$ kernels in every $4^{\text{th}}$ layer and $1 \times 1$ (i.e., pointwise linear combinations across all channels) for the rest, resulting in a total of $214,144$ parameters. For comparison, we also trained conventional (non-multi-scale) CNNs for denoising. For RF $43 \times 43$, we used the same architecture as for the coarsest scale band of the multi-scale denoiser: 21 bias-free convolutional layers with no subsampling. To create smaller RFs, we followed the same strategy of setting some filter sizes in the intermediate layer to $1 \times 1$.

**Training**. For experiments shown in Figure 3 and Figure 4, we use $202,499$ training and 100 test images of resolution $160 \times 160$ from the CelebA dataset (Liu et al., 2015). For experiments shown in Figure 5, Figure 7 and Figure 6, we use $29,900$ train and 100 test images, drawn from the CelebA HQ dataset (Karras et al., 2018) at $320 \times 320$ resolution. We follow the training procedure described in (Mohan* et al., 2020), minimizing the mean squared error in denoisingd images corrupted by i.i.d. Gaussian noise with standard deviations drawn from the range [0, 1] (relative to image intensity range [0, 1]). Training is carried out on batches of size 512. Note that all denoisers are universal and blind: they are trained to handle a range of noise, and the noise level is not provided as input to the denoiser. These properties are exploited by the sampling algorithm, which can operate without manual specification of the step size schedule Kadkhodaie & Simoncelli (2021).

## D  WAVELET CONDITIONAL SYNTHESIS ALGORITHM

Sampling from both the CNN and cCNN denoisers is achieved using a slightly modified version of the algorithm of **?**, as defined in Algorithm 1. This method uses only two hyperparameters, aside from initial and final noise levels, and their settings are more forgiving than those of backward SDE discretization parameters in score-based diffusions. A step size parameter, $h \in [0, 1]$, controls the trade-off between computational efficiency and visual quality. A stochasticity parameter, $\beta \in (0, 1]$, controls the amount of noise injected during the gradient ascent. For the examples in Figure 5, Figure 7 and Figure 6, we chose $h = 0.01$, $\sigma_0 = 1$, $\beta = 0.1$ and $\sigma_\infty = 0.01$.

Image synthesis is initialized with a terminal low-pass image (either sampled from the associated CNN, or computed from a test image), and successively sampling from the wavelet conditional distributions at each scale, as defined in Algorithm 2.

---

**Algorithm 1** Sampling via ascent of the log-likelihood gradient from a denoiser residual

---

**Require:**  denoiser $f$, step size $h$, initial noise level $\sigma_0$, final noise level $\sigma_\infty$
  1: $t = 0$
  2: Draw $\boldsymbol{x}_0 \sim \mathcal{N}(0, \sigma_0^2 \mathrm{Id})$
  3: **while** $\sigma_t \geq \sigma_\infty$ **do**
  4:      $t \leftarrow t + 1$
  5:      $\boldsymbol{d}_t \leftarrow f(\boldsymbol{x}_{t-1}) - \boldsymbol{x}_{t-1}$                  $\triangleright$ Compute the score from the denoiser residual
  6:      $\sigma_t^2 \leftarrow \|d_t\|^2 / N$             $\triangleright$ Compute the current noise level for stopping criterion
  7:      $\gamma_t^2 = \left((1 - \beta h)^2 - (1 - h)^2\right) \sigma_t^2$
  8:      Draw $z_t \sim \mathcal{N}(0, I)$
  9:      $\boldsymbol{x}_t \leftarrow \boldsymbol{x}_{t-1} + h\boldsymbol{d}_t + \gamma_t z_t$ $\triangleright$ Perform a partial denoiser step to remove a fraction of the noise
10: **end while**
11: **return** $\boldsymbol{x}_t$

---

**Algorithm 2** Wavelet Conditional Synthesis

---

**Require:**  number of scales $J$, low-pass image $\boldsymbol{x}_J$, conditional denoisers $(f_j)_{1 \leq j \leq J}$, step size $h$, initial noise level $\sigma_0$, final noise level $\sigma_\infty$
  1: **for** $j \in \{J, \ldots, 1\}$ **do**
  2:      $\bar{\boldsymbol{x}}_j \leftarrow \mathrm{DrawSample}(f_j(\cdot, \boldsymbol{x}_j), h, \sigma_0, \sigma_\infty)$         $\triangleright$ Wavelet conditional sampling
  3:      $\boldsymbol{x}_{j-1} \leftarrow W^T(\bar{\boldsymbol{x}}_j, \boldsymbol{x}_j)$                $\triangleright$ Wavelet reconstruction
  4: **end for**
  5: **return** $\boldsymbol{x}_0$

---

## E  ADDITIONAL SUPER-RESOLUTION EXAMPLES

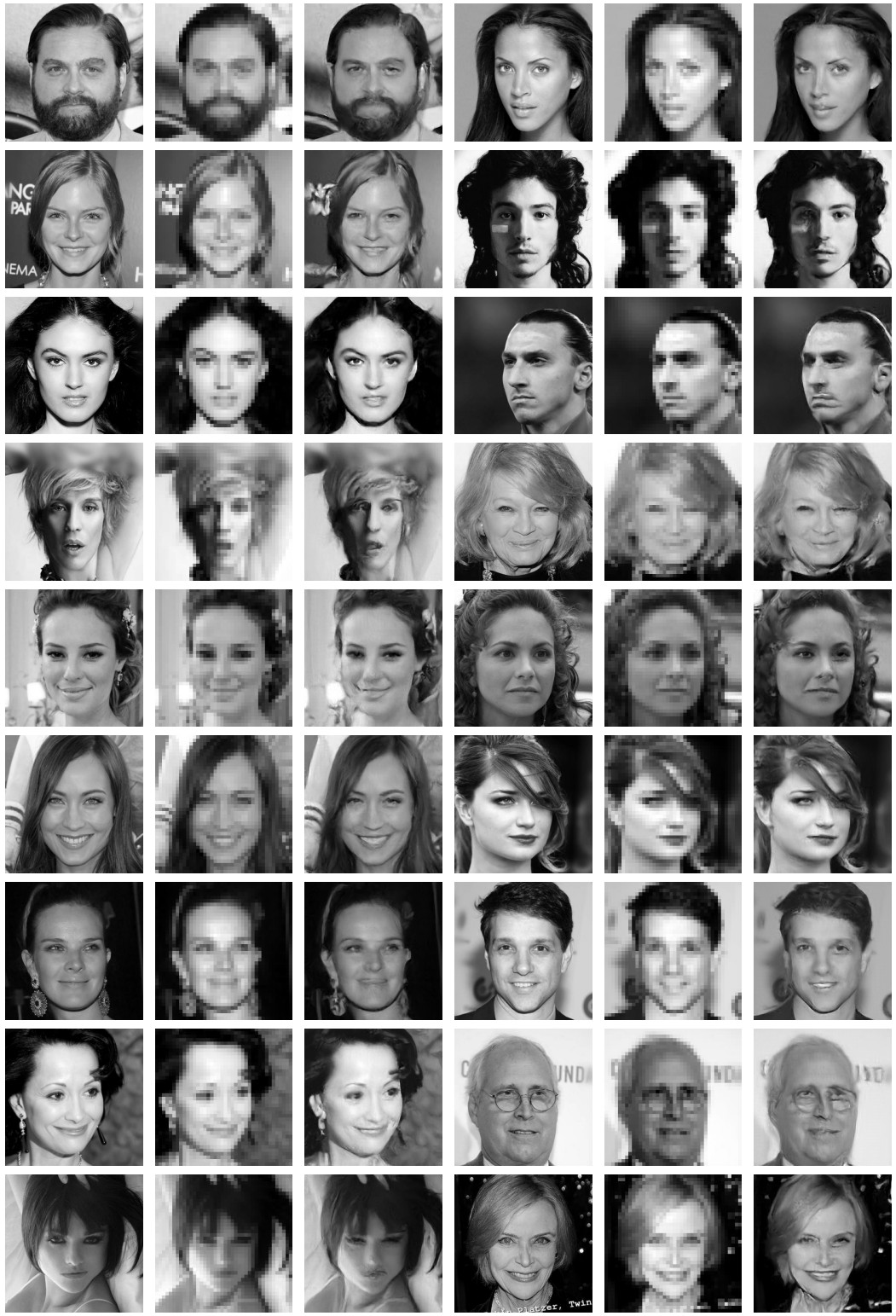

Figure 7: Additional super-resolution examples. **Columns, in groups of three, from left to right:** original images, $320 \times 320$ pixels. Low-pass images of corresponding 3-stage wavelet decomposition (downsampled to $40 \times 40$, expanded to full size for viewing). Coarse-to-fine conditional generation using the multi-scale conditional denoisers, each with RF size $13 \times 13$.

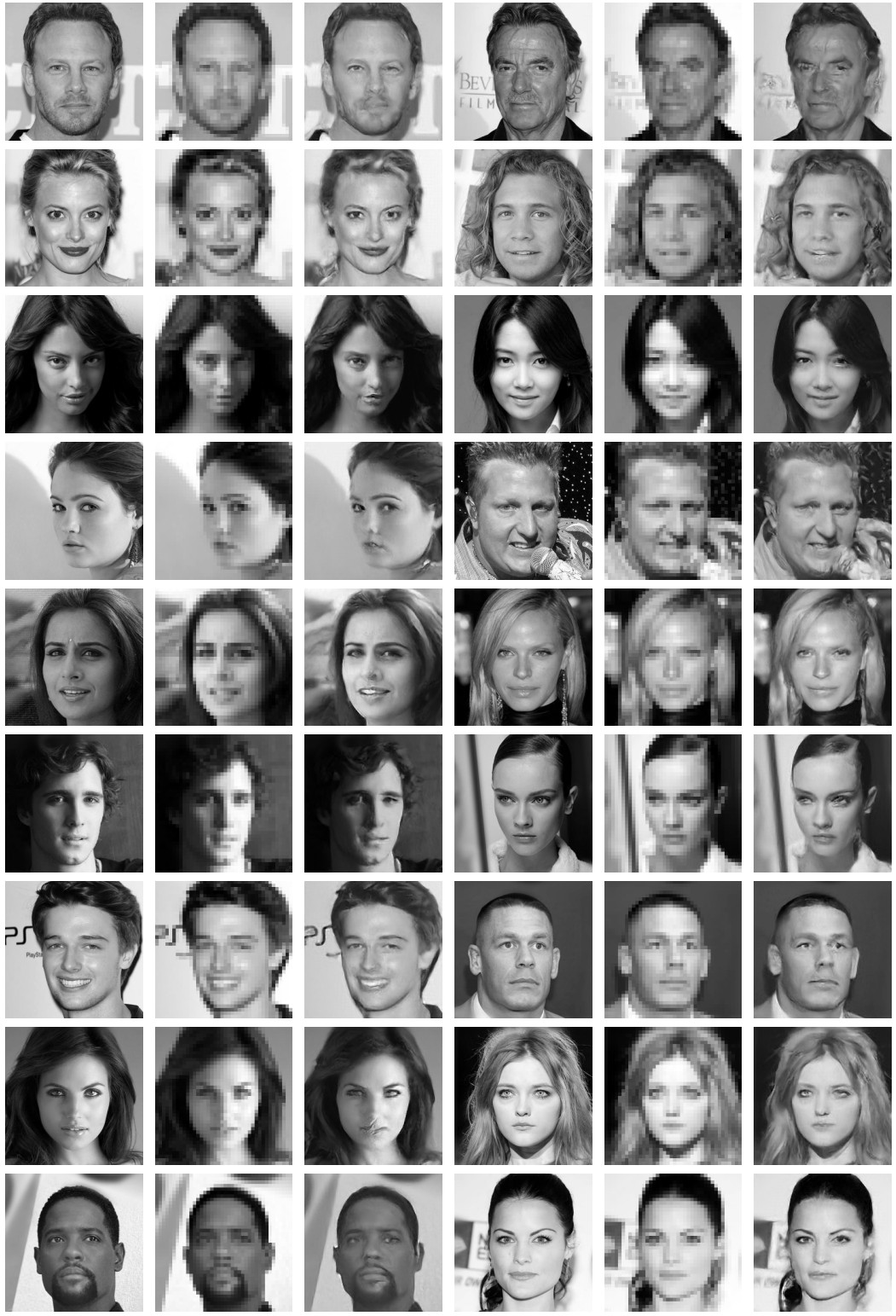

Figure 8: Additional super-resolution examples. See caption of Fig. 7.

