# OpenReview forum: "Learning multi-scale local conditional probability models of images"
_ICLR.cc/2023/Conference — ICLR 2023 notable top 25%_

### Official Review · Reviewer_KhB3 · 2022-10-23

**Confidence:** 4
**Correctness:** 4
**Technical Novelty And Significance:** 4
**Empirical Novelty And Significance:** 3
**Recommendation:** 10

**Clarity, Quality, Novelty And Reproducibility:**

Clarity: the writing/math is clear and interesting
Quality and novelty: high because it makes explicit models that are regularly hidden
Reproducibility: I missed links to code

**Strength And Weaknesses:**

STRENGTHS:

The paper should be accepted because it builds an explicit image probability model as opposed to implicit (non-accessible) models of autoencoders, GANs, normalizing flows, or diffusion methods. These results are an interesting connection between the efforts of (1) linking the gradient of the PDFs with optimal denoising [Miyasawa61, Raphan&Simoncelli11, Vincent11], and (2) establishing conditional relations between low-frequency and high-frequency aspects in spatial neighborhoods of natural images [Buccigrossi&Simoncelli99, Marchand et al.22].

WEAKNESSES:

* The model reminds me of other recent conditional models such as the PixelCNN [van den Oord et al.16, Salimans et al.17].
I guess the key difference is the formulation in a multiscale representation instead of in the input spatial domain.
The authors already compare with CNN models in the spatial domain. I think it would be interesting for the reader to see some example with more sophisticated conditional models in the spatial domain such as the PixelCNN. Of course, in this comparison the size of the images should be big enough for a couple of scales but small enough for proper training of pixelCNN.

* The authors acknowledge the main limitation of their proposal in the discussion "the non-stationary capabilities of the full model arise primarily from the terminal low-pass CNN, which uses global receptivie fields". I think this should be acknowledged from the very beginning (in the abstract and/or intro) because the interesting conditional relations (the core of the proposal) always need "a seed to grow" which is modeled in a conventional way.

* The dimensionality of the model is small because the Markov structure is spatially local (the interesting theorem 1).
Could we get extra reductions in the parameters using locality in the orientation and in the scale in redundant wavelets?
Note that nonorthonormal wavelets could also be applied since it would only imply the consideration of the determinant |W| in Eq. 1, right?. This would mean the extension to orientation and scale of the locality concept applied in space.

REFERENCES:

* van den Oord et al. 16a, Salimans et al.17 Conditional Image Generation with PixelCNN Decoders NeurIPS 2016 arxiv 1606.05328

* van den Oord et al. 16b Pixel Recurrent Neural Networks ICML 2016 arxiv 1601.06759

* Salimans et al.17 PixelCNN++: Improving the PixelCNN with discretized logistic mixture likelihood and other modifications ICLR arxiv 1701.05517

MINOR COMMENTS:

* You could cite Marchand et al. 22 the first time you mention that the conditional can be modeled with conditional Gibbs energies (Eq. 2)

* The first paragraph of section 5 mentiones the super-resolution problem and it does not mention the synthesis problem, but then, the next paragraphs alternate between one problem and the other. The two problems should be mentioned at the begining to avoid confusion.

* The proof of Eq. 5 (appendix B) already was done in Raphan&Simoncelli 11, right?. Why not just cite that?

**Summary Of The Paper:**

The authors propose a local conditional probability model for images in multiscale representations.
In this model, the probability of the high-frequency (detail) components is conditioned to the low-frequency (low-resolution) component, and these conditional relations are spatially local. The probability of the final low-resolution residual is modeled globally.

Both the conditional probabilities and the probability of the low-resolution residual are learnt by CNN denoisers using the relation between the denoiser errors and the gradient of the probability of the signal.

The local (Markov) structure of the conditional probabilities means that the parameters (receptive fields) of the CNNs are restricted to the local neighborhood and hence the conditional part of the model is relatively low dimensional. This assumes that the conditional relations at each scale are essentially stationary (or translation invariant).

However, the receptive fields of the CNN to model the probability of the low-pass residual have to be the same size as the residual in order to capture its non-stationary nature. The requirement of dense connection for the low-pass residual is the main limitation of the proposed model in terms of dimensionality.

**Summary Of The Review:**

The paper should be accepted because it builds an explicit image probability model as opposed to implicit (non-accessible) models of autoencoders, GANs, normalizing flows, or diffusion methods. These results are an interesting connection between the efforts of (1) linking the gradient of the PDFs with optimal denoising [Miyasawa61, Raphan&Simoncelli11, Vincent11], and (2) establishing conditional relations between low-frequency and high-frequency aspects in spatial neighborhoods of natural images [Buccigrossi&Simoncelli99, Marchand et al.22]. Experimental results successfully show the good performance of the proposed model.

However, some elaboration would be appreciated on: (1) advantages over alternative conditional models (such as PixelCNN) in experiments, (2) acknowledgement of the central role of the low-pass component (not a core of the proposal), (3) possible extensions of the locality concept in space to orientation and scale.

My current score is 7, but I will be happy to raise to 8 or 9 it if the authors address the suggested points.

---

> ### Author Response · Authors · 2022-11-19
> **Official Response to Reviewer KhB3**
>
> We thank the reviewer for their helpful comments. Regarding the three concerns raised:
>
> - Pixel-CNN and follow-ups are another class of conditional model, but their conditioning is global (and as with other models operating in the pixel domain, this is critical for their success). Our goal was to examine dimensionality reduction of the model by imposing locality in the conditional probabilities, and to examine empirically how small the conditioning neighborhoods can be. As a result, we have not compared our model to PixelCNN or any other global models. One can modify the PixelCNN architecture to restrict its RF, but that would be analogous to restricting the RF of a global DnCNN, which results in a poor model of images, as we have demonstrated.
>
> - The reviewer is correct that the terminal CNN on the small low-pass images plays an important role in the model, and we were negligent in not explaining that early in the text. We now make this important point in the Abstract and Introduction, as well as when defining the model. We demonstrate that the global and non-stationary structures are captured by the CNN with an RF that covers the entire low-pass image. This non-stationary low-pass model provides a starting point for the conditional models, which produces high-resolution images which are non-stationary, with large-scale structures. The low-pass model is indeed a limitation in the sense that its size currently dominates the model dimensionality, yet it remains bounded by the small image size. Understanding whether it is possible to further reduce this low-pass model is an important direction for future research.
>
> - Extensions of the method to locality over orientation and scale is an interesting idea, and we have discussed these options as the work was being developed. We decided to focus iniitially  on spatial extent, which is most effective in reducing dimensionality, and seemed likely to have the most potent effect on performance.  In practice, there are many generalized neighborhood configurations one could consisder, and it is not clear how to choose.  A potential direction is to look for neighbors with largest mutual information, or the largest values in the precision matrix. In any case, it's an open direction for our future work. With regard to spatial extent, we have now augmented the results in our original submission by including 9x9 RF cCNNs, which preserve high-quality results (updated Fig 3)  and reduces model dimensionality (relative to the 13x13 RF) by roughly a factor of two.
>
> We have clarified the relationship to the previous work of Marchand et al. 22 and Raphan & Simoncelli 11, and modified the text to express the distinction between the super-resolution and synthesis results more clearly.
>
> We will release the code to reproduce the paper’s results with an appropriate pointer in the camera-ready version, to guarantee the reproducibility of all numerical experiments.

---

> > ### Comment · Reviewer_KhB3 · 2022-12-10
> > **I rise my score. Accept!**
> >
> > Your answers on the lack of comparisons mentioning the sizes of the required receptive fields convinced me.
> > Authors should ensure that these arguments (expressed in different ways for the different reviewers -one way or another-) should be present and clear in the final version of the work. Somehow I missed them in my first reading and then I took the "easy" criticism: *I want to see an empirical comparison*...
> >
> > I think this is an important paper because (1) it is a very explicit and understandable model [as opposed to the conventional dark-box-brute-force-dense-nets models], and (2) it builds on an image property (inter-scale predictability) which helps to reduce the analysis problem to the "yet-to-be-understood" long-range structure that lives in the low-pass residual of images.
> >
> > Definitely, I think the paper has to be accepted.
> >
> > To do so I rise my score from 7 to 9.

---

> > > ### Author Response · Authors · 2022-12-11
> > > **Thank you**
> > >
> > > Thank you for your encouraging vote of confidence - we too are enthusiastic about the results! We will make sure that our point about lack of comparisons with other models is clear in the final version of the paper.
> > >
> > > We also noticed that the recommended score in OpenReview is not updated to the new score. To avoid confusions when it comes to the final decision, we wanted to respectfully ask if you could update the score.

---

### Official Review · Reviewer_Pfsy · 2022-10-25

**Confidence:** 3
**Correctness:** 3
**Technical Novelty And Significance:** 3
**Empirical Novelty And Significance:** 3
**Recommendation:** 8

**Clarity, Quality, Novelty And Reproducibility:**

The paper needs improvement in clarity and organization. For example, it would be nice if in terms of bullet points the authors can indicate the contributions of this paper and how it differs from prior works, and how it benefits the ICLR community. The method is novel in how it combines ideas to scale up wavelets.

**Strength And Weaknesses:**

The paper shows the application of their method in several image tasks (generation, denoising, super-resolution).

The paper can improve on clarity. It is not clear what their contribution is from a general perspective on the performed tasks (e.g., denoising and super-resolution, generation). Please explain the advantages of your method compared to other general frameworks used for super-resolution, denoising, etc. It is not shown how their framework performed against multiple baselines. There seems to be only one method that they compared with which is the vanilla version of their Markov method. Here are detailed comments.

Lack of sufficient baselines:
- By using local convolutional filters and learning locally, their method alleviates the curse of dimensionality; However, how is their method compared to other popular methods in the literature in terms of performance, computational complexity, runtime, etc? (Perhaps, a quantitative comparison between their method and the cited methods on the last paragraph of page 8 is suggested)
- It is nice to see (in Figure 3) that their method outperforms a similar approach of Markov probability in the pixel domain. However, more experiments with non-Markov baselines should be included. How is the performance of this method for example compared to generic deep learning (e.g., DnCNN bias-free) for denoising, super-resolution, etc?

The literature review needs proper citations. Here are some examples in the introduction. Please provide citations to
- the second line in the intro talks about dimensionality.
- the fourth line in the intro refers to traditional methods.
- the last line of the first paragraph in the intro refers to global models.

It is hard to see the relationship between the conditional CNN and (3). Can the authors explain?

A thorough experimental analysis would be nice to add to the characterization of their method: how the performance changes as the number of wavelets, and the depth of scale change.

It is not clear if the models are trained on the same noise level compared to the test noise or a range. Please clarify and add such information.

How is the performance on natural images?

It is not clear how the right image in Figure 6 is generated. Please elaborate on "CNN denoiser with TF smaller than the image".

Minor comments

- Moving Figure 2 to the second page will help to understand the - model.

- Why the first row of Figure 4 is smaller? Please fix.

- Elaborate on the statement after Figure 4: "... less than three percent of the coefficients".

**Summary Of The Paper:**

The paper introduces a multi-scale conditional probability model to reconstruct and denoise images. Specifically, the scales come from wavelet transform, and they generalize Markov conditional models by parametrizing the conditional gradients with a CNN with a local receptive field. They show how their method performs better than Markov methods conditioned and performed in the pixel domain. They study the effect of the receptive field. They use a previously proposed method to generate new images, i.e., to perform gradient ascent on the log probability of the learned CNNs to draw samples from the learned image distributions. They show applications in denoising, synthesis, and super-resolution.


**Summary Of The Review:**

The paper proposes a method to generalize multiscale wavelets and markov models to learn conditional distributions of wavelets to recover images. The method needs a thorough experimental study to show its performance against other learning frameworks and to clearly explain how its method differs from prior works and its advantages. Hence, the paper in its current form is not recommended for acceptance.


-----------

 Given the authors' responses, I have raised my score to 8.

---

> ### Author Response · Authors · 2022-11-19
> **Official Response to Reviewer Pfsy**
>
> We thank the reviewer for their helpful comments.
>
> We have rewritten the Abstract, Introduction, and portions of the Discussion to clarify our approach, its relationship to other literature, and our contributions. The goal of the paper is not to introduce a new algorithm to generate images or perform super-resolution but to explain why score diffusion models can do it so well. The main surprise of these algorithms is the ability of neural networks to approximate scores of high-dimensional distributions, despite the curse of dimensionality. State-of-the-art algorithms use cascaded diffusion to estimate the score, with no underlying model or mathematical justification. To the best of our knowledge, our paper is the first to provide an explicit mathematical model explaining why the score can be estimated without suffering from the curse of dimensionality, thanks to highly localized Markov properties across scales.
>
> The reviewer mentions the lack of comparison with state-of-the-art methods on denoising and super-resolution. This is intentional because the main goal was to demonstrate the existence of Markov conditional properties across scales but not in the pixel domain. For doing so, we used small size CNNs which have about 0.001 of parameters of SoTA architectures (i.e., on the order of hundreds of thousands as opposed to hundreds of millions or billions), but whose receptive fields could be modified in order to demonstrate these local Markov properties. We do however compare with a generic bias-free DnCNN in Figure 3 (“RF = 43x43” curve in left panel), which is now more clearly explained in the caption. Comparisons with state of the art algorithms are beyond the scope of the paper and would require using networks of similar size. However, we have now verified that our multiscale Markov model performs much better than pixel-domain Markov models for more complex images such as bedroom images of the LSUN dataset (Yu et al 2015).
>
> Following the reviewer remarks, we improved the clarity of the paper. Regarding the reviewer’s specific questions:
> - Eq. (3) represents the target function that is approximated by the conditional CNN. The CNN thus takes as input $\bar x_j$ together with $x_j$ and returns an approximation of the conditional score $\nabla_{\bar x_j} \log p(\bar x_j | x_j)$.
> - Figure 3 evaluates the denoising performance across a wide noise level range: the x axis represents the PSNR between the input noisy image and clean ground truth. The networks are similarly trained on a noise level range: the noise standard deviation is sampled with a density $p(\sigma) \propto 1/\sqrt{\sigma}$ where $\sigma$ ranges from $0$ to $1$ (that corresponds to PSNR $\infty$ to $0$). We have added more details on training to Appendix C.
> - The “less than 3%” referred to the fraction of coefficients in the conditioning window relative to the entire image in the finest scale (13x13 / 80x80). We now updated that to 1% for a 9x9 RF cCNN.
> - We corrected some typos and awkward sentence issues, and improved captions of figure 3 and 6.
>
> The reviewer rates the empirical novelty as marginally significant. We respectfully disagree. This paper demonstrates empirically that image probability distributions have conditional Markov properties across scales, on small neighborhoods, which was unknown. Moreover, it shows how score diffusion algorithms can take advantage of such properties for generation, denoising and super-resolution. With a complementary experiment, we now demonstrate that conditional neighborhood sizes can be reduced to 9x9 spatial areas which is quite spectacular.

---

> > ### Comment · Reviewer_Pfsy · 2022-12-06
> > **Reviewer's Comments after Reading Authors' Response**
> >
> > I thank the authors for their responses. I have read other reviews and authors' responses. The authors have addressed the majority of my comments; this has improved clarity. Authors have made it clear that their novelty is in building an explicit model of images which can result in efficient image generation, denoising, etc. compared to other implicit methods such as generic CNNs. Given the authors' argument on their novelty, I have increased my overall and empirical novelty scores.
> >
> > Still, the authors' reservations to compare with certain methods in the literature are not fully understood. Every method has its own advantages and disadvantages. The authors have mainly pointed out their advantages, but do not provide a performance comparison to other conditional methods or certain state-of-the-art (e.g., Pixel-CNN, pointed out by other reviewers). I see why their proposed method may not be comparable to some baselines only in performance (as they offer efficient scalability). However, I still think that including such comparisons would be useful. I strongly recommend the authors include such quantitative comparisons, but argue in the paper about their trade-offs such as efficiency and sample and model complexity.

---

> > > ### Author Response · Authors · 2022-12-11
> > > **Thanks**
> > >
> > > Thank you for raising the score. We are glad our updates successfully addressed your comments and clarified some ambiguities.
> > >
> > > Regarding comparisons to state of the art diffusion models, we still believe that including those might result in more confusion about the point of our paper. State of the art score based diffusion models, as well as Pixel-CNN, have effectively global receptive fields. So it is not clear what conclusions one can draw from such comparisons. We will make sure that our point regarding comparisons to SOTA models is clear in the final version.

---

### Official Review · Reviewer_CgEx · 2022-10-27

**Confidence:** 2
**Correctness:** 4
**Technical Novelty And Significance:** 3
**Empirical Novelty And Significance:** 3
**Recommendation:** 8

**Clarity, Quality, Novelty And Reproducibility:**

The paper has its novelty. The quality of the paper presentation is very good. The paper is reproducible.

**Details Of Ethics Concerns:**

The source of the face dataset has not been mentioned. It is not clear whether human ethics has been approved or not.

**Strength And Weaknesses:**

Strength:

- This work is well-motivated.
- This paper addresses an important problem in learning prior probability models of images.
- The paper seems to be solid.
- Two application scenarios are provided with technical details.

Cons:
- The image reconstruction results can be further improved.
- There is no comparison with the state-of-the-art methods on the two mentioned tasks.

**Summary Of The Paper:**

This paper generalizes a Markov wavelet conditional probability model developed based on the renormalization group theory. The model is parameterized using conditional CNNs with local receptive fields, which enables stationary Markov properties. The method is extended to handle several face image reconstruction tasks, e.g. image denoising and synthesis. The results show that the model can capture image features with a small receptive field,

**Summary Of The Review:**

This is a well-written paper with solid theoretical content. The experiments are insufficient. Given the theoretical nature of the paper, this can be somehow tolerated.

The authors' responses have well addressed by concerns. I have raised the empirical novelty and significant score and also raised the final recommendation to 8.

---

> ### Author Response · Authors · 2022-11-19
> **Official Response to Reviewer CgEx**
>
> We thank the reviewer for their helpful comments.
>
> The main concern is the lack of comparison with state-of-the-art methods on denoising and super-resolution. This is intentional, as mentioned (but perhaps insufficiently explained) in the last paragraph before the Discussion. We find the ability of denoising networks to approximate scores of high-dimensional distributions remarkable (shocking, really!), given the curse of dimensionality. The state-of-the-art algorithms use coarse-to-fine cascaded steps, but they do not offer a justification or connection to any underlying model or mathematical property. Moreover, they use U-Net architectures with global receptive fields that are not convolutional, with huge parameterizations (hundreds of millions, compared to ours which uses hundreds of thousands). The goal of our paper is not to improve performance, but to explain and demonstrate that the reason the score can be estimated this way is because image probabilities have highly localized *conditional Markov properties* across scales. To demonstrate this, we used a generic convolutional architecture (no downsampling/upsampling) to ensure stationarity, and to allow easy modification of the conditional neighborhood size. Enforcing these two properties results in low dimensionality, which is independent of image size. Our architecture also allowed us to demonstrate the absence of Markov properties in the pixel domain. To the best of our knowledge, this is the first paper that attempts to explain why the score (and corresponding density) can be estimated without suffering from the curse of dimensionality, and why cascaded diffusion is essential to generate large size images.
>
> We also respectfully disagree with the rating indicating marginal significance of empirical novelty.  All existing score diffusion methods compute estimates of the score by training parametric models with enormous complexity and very little interpretability. In this paper, we demonstrate that image probability distributions have conditional Markov properties across scales, on small neighborhoods, and show how score diffusion algorithms can take advantage of such properties. This is an empirical result, since it is not known a priori, and cannot be obtained purely from theoretical considerations, and we are not aware of any prior work that demonstrates these properties. We have now added an additional comparison (updated figure 3) showing that conditional neighborhood sizes can be reduced to 9x9 spatial regions - quite spectacular, when compared with global state-of-the-art models!
>
> We have rewritten the Abstract, Introduction, and portions of the Discussion to clarify our approach, its relationship to other literature, and our contributions.
>
> Regarding ethical concerns, we state that the models are trained on the CelebA and CelebA-HQ datasets, which are standard public face datasets. Due to lack of diversity in these datasets, we recommend that they be restricted to academic use only until more diverse datasets are curated.

---

### Decision · Program_Chairs · 2023-01-20

**Decision:**

Accept: notable-top-25%

**Justification For Why Not Higher Score:**

The main weakness of the submission, which was pointed out by *all* reviewers, and I agree, was disagreed with by the authors: that the method does not sufficiently compare to SOTA methods empirically.  The authors argue that other models have orders of magnitude more parameters and that it would be confusing to make the comparison.  I think the authors need to trust the readers more.  I understand the frustration of reviewers not getting the point and viewing everything as a benchmark competition where only accuracy matters, but clearly that did not happen in this review process.  The authors should trust that, with an appropriate presentation, e.g. some notion of accuracy vs. model size as a scatter plot, the ICLR readership is capable of discerning this fact.  The reader is now left with the additional task of either providing their own benchmarking, or in searching across different papers with the understanding that the evaluation setup may not be consistent.  Why???

**Justification For Why Not Lower Score:**

Due to the above concern, part of me even wanted to reject the paper for lack of benchmarking - a common concern across *all* reviewers.  The authors response is not really satisfactory.  As such, I'm OK with this being bumped down.  On the other hand, the reviewer scores are quite high, and the main point of the paper is well conveyed and demonstrated - that a frequency domain Markov property is a surprisingly effective image model, with eventually lessons for design and understanding of SOTA models as well as being a very interesting model in its own right.

**Metareview: Summary, Strengths And Weaknesses:**

The submission proposes an image model based on a wavelet framework.  This is quite similar to other works that attempt to understand the behavior of NNs in the frequency domain using multiple frequency transforms.  The main thing that sets it apart is the imposition of a Markov condition that severely theoretically restricts the model over previous families of solutions.  Quite interestingly, this results in quite performant models at a tiny fraction of the number of parameters.  This is an interesting family of models to study and refine in its own right, but it also gives insights into the performance of other coarse to fine strategies, which may be viewed as more heuristic or pragmatic.  The exposition is clear and technically competent, and the contributions of the paper were unanimously appreciated by the reviewers with Reviewer KhB3 recommending that it be highlighted in the conference.

The main weakness of the submission is the dismissal of the authors of multiple requests for additional empirical comparisons to SOTA models.  The baseline model is essentially an ablated version of the proposed model, and does not remotely reach the theoretical capacity of the current SOTA, which was requested by all reviewers.  The author response was
> The reviewer mentions the lack of comparison with state-of-the-art methods on denoising and super-resolution. This is intentional because the main goal was to demonstrate the existence of Markov conditional properties across scales but not in the pixel domain. For doing so, we used small size CNNs which have about 0.001 of parameters of SoTA architectures....

The point is somewhat taken, but that does not preclude including that information.  While the paper is interesting, there is still clearly a gap between the models evaluated in the paper and the existing literature.  A scatter plot of model size vs. accuracy could be a way to address this, and certainly a paragraph in the Discussion section prominently highlighting these limitations should be included.  As it stands, the authors are indicating that they intentionally left additional benchmarking and scaling of their family of models to future work, which is not a very compelling answer.

**Note From Pc:**

if the above contains the word "oral" or "spotlight" please see: "oral" presentation means -> notable-top-5% and "spotlight" means -> notable-top-25%. As stated in our emails, we are disassociating presentation type from AC recommendations